Long-term survival, growth, and reproduction of Acropora palmata sexual recruits outplanted onto Mexican Caribbean reefs

Mendoza Quiroz Sandra 1 2
Beltrán-Torres Aurora Urania 3
Grosso-Becerra Maria Victoria 2
Muñoz Villareal Daniela 4
Tecalco Rentería Raúl 1 2
Banaszak Anastazia T. banaszak@cmarl.unam.mx 2
1 SECORE International , Miami , FL , United States of America
2 Unidad Académica de Sistemas Arrecifales, Universidad Nacional Autónoma de México , Puerto Morelos , Quintana Roo , Mexico
3 Jardín Botánico, El Colegio de la Frontera Sur , Puerto Morelos , Quintana Roo , Mexico
4 Universidad Jorge Tadeo Lozano , Santa Marta , Magdalena , Colombia
Kamel Bishoy
Electronic publication date: 2023 Aug 1
Publication date: 2023
Volume: 11
Electronic Location ID: e15813
Received 2023 Jan 18; Accepted 2023 Jul 7
Copyright: ©2023 Mendoza Quiroz et al.
Copyright year: 2023
Copyright holder: Mendoza Quiroz et al.
License: This is an open access article distributed under the terms of the Creative Commons Attribution License, which permits unrestricted use, distribution, reproduction and adaptation in any medium and for any purpose provided that it is properly attributed. For attribution, the original author(s), title, publication source (PeerJ) and either DOI or URL of the article must be cited.
License URL: https://creativecommons.org/licenses/by/4.0/

Keywords: Ecological restoration, Reef-building corals, Coral spawning, Coral reproduction, Coral breeding, Larval propagation

Funding: Comisión Nacional para el Conocimiento y Uso de la Biodiversidad JA009 Universidad Nacional Autónoma de México Consejo Nacional de Ciencia y Tecnología 247765 This work was supported by the Comisión Nacional para el Conocimiento y Uso de la Biodiversidad (No. JA009), a Universidad Nacional Autónoma de México grant to ATB, and the Consejo Nacional de Ciencia y Tecnología (No. 247765) to ATB. The funders had no role in study design, data collection and analysis, decision to publish, or preparation of the manuscript.

==============================
Acropora palmata is a foundational yet endangered Caribbean reef-building coral species. The lack of recovery after a disease outbreak and low recruitment has led to widespread use of fragmentation to restore populations. Another option is the production of sexual recruits (settlers) via assisted reproduction to improve the genetic diversity of depleted populations; however, the viability of this approach has not been tested over the long term. In 2011 and 2012, A. palmata larvae were cultured, settled, and the sexual recruits raised in an ex-situ nursery. Survival and growth were monitored over time. In 2014, these two F1 cohorts were moved to an in-situ nursery and after one year, a subset (29 colonies) was outplanted onto Cuevones Reef in the Mexican Caribbean. Growth and survival of these colonies were monitored periodically and compared to colonies that remained in the in-situ nursery. In 2019, samples were collected and analyzed for fertility and fecundity. 53% of the colonies were gravid and fecundity was 5.61 ± 1.91 oocytes and 3.04 ± 0.26 spermaries per polyp. A further 14 colonies from these two cohorts were outplanted in 2020 onto Picudas Reef and monitored during the subsequent spawning seasons. Two years after outplanting onto Picudas Reef, all colonies were alive and spawning of three of these colonies was recorded in 2022 in synchrony with the wild population. Gametes were collected from two colonies and crossed, with 15% fertilization success. Spermatozoa from wild colonies were then added and fertilization success increased to 95%. The resultant larvae followed normal development and symbiont uptake was visible within two weeks. The F2 generation was settled, maintained in an ex-situ nursery, and monitored for survival and growth. Both F1 and F2 generations followed a Type III survival curve with high initial mortality while in the ex-situ nursery and low later-stage mortality. The growth rates of these colonies increased three-fold after outplanting when compared to their growth rates in the ex-situ and in-situ nurseries. All colonies survived while in the in-situ nursery and for an additional nine years after outplanting onto Cuevones Reef. Overall, our results show that colonies produced by assisted breeding, once outplanted, may contribute to the genetic diversity and establishment of self-sustaining sexually-reproducing populations, which is an overarching goal of coral restoration programs.

Introduction

A long-term trend in declining coral cover has been documented in the Caribbean (Gardner et al., 2003; Cramer et al., 2020). Since the 1980s, reefs along the Mexican Caribbean have shown signs of degradation due to a combination of disturbances (Gómez et al., 2022). Acropora palmata is a foundational Caribbean reef-building species and was abundant, forming reef crests in the Mexican Caribbean prior to the 1980s (Jordán-Dahlgren & Rodríguez-Martínez, 2003). However, currently it is listed as critically endangered on the IUCN Red List (International Union for the Conservation of Nature, 2023).

During the 1980s, white-band disease was the primary cause of the greater than 95% decline in A. palmata abundance in the Caribbean (Gladfelter, 1982; Aronson & Precht, 2001; Precht et al., 2002). Unabating anthropogenic disturbances such as the marked increase in coastal development due to the ever-expanding tourism market and associated urbanization (Rioja-Nieto & Álvarez Filip, 2019; Häder et al., 2020; Banaszak, 2021) combined with hurricanes (Jordán-Dahlgren & Rodríguez-Martínez, 1998; Gómez et al., 2022) and sea surface thermal anomalies (Bove, Mudge & Bruno, 2022) threaten the viability of this and many other coral species (Miller, Weil & Szmant, 2000; Rodríguez-Martínez, Banaszak & Jordán-Dahlgren, 2001; Vermeij et al., 2011; Rodríguez-Martínez et al., 2014; Richmond, Tisthammer & Spies, 2018; Alvarez-Filip et al., 2019; González-Barrios, Cabral-Tena & Alvarez-Filip, 2021).

One example of such degradation is Picudas Reef located in the Puerto Morelos Reef National Park in the Mexican Caribbean. Monitoring of this reef has shown that it has a 3% prevalence of both White Pox and White Band disease, high cyanobacterial and macroalgal cover as well as small A. palmata colonies (Banaszak & Álvarez Filip, 2014). Artisanal fishing is allowed on this reef, which is rarely visited by tourists (Instituto Nacional de Ecología, 2000). In recent years the corals on this reef have been severely impacted by the outbreak of Scleractinian Coral Tissue Loss Disease (Guzmán-Urieta & Jordán-Dahlgren, 2021).

Reefs can also be damaged by ships. In December 1997, a cruise ship impacted Cuevones Reef located in the Isla Mujeres, Punta Cancún, and Punta Nizuc National Park, primarily affecting the reef-building species Acropora palmata growing on the reef crest (Victoria-Salazar et al., 2023). This grounding affected a total area of 504 m2 reducing the live coral cover from 14% to 1%. By 2016, coral cover had recovered to 4.5% principally composed of Porites astreoides and Agaricia agaricites, but there was no evidence of natural recruitment of reef-building species such as A. palmata (Victoria-Salazar et al., 2023).

Acropora palmata has yet to recover to its historical highs on the reefs in the Mexican Caribbean (Rodríguez-Martínez et al., 2014), which has led to widespread interventions to replenish depleted populations, particularly using coral fragments (Bayraktarov et al., 2020). The assisted production of sexual recruits is also important for population replenishment to maintain or potentially increase genetic diversity and enable adaptation to changing environmental stressors (Baums, 2008; Baums et al., 2013) as well as to improve population connectivity (Baums et al., 2013; Chamberland et al., 2015; Baums et al., 2019).

Acropora palmata is a hermaphrodite, housing oocytes and spermaries in the same polyp (Szmant, 1986). Spawning of oocytes and sperm, packaged together in gamete bundles, occurs in the summer months after the full moon, typically in August (Szmant, 1986; Jordan, 2018; Fogarty & Marhaver, 2019). Acropora palmata colonies have been reared from larvae, outplanted and shown to be reproductively active at four years of age (Chamberland et al., 2015; Chamberland et al., 2016). A knowledge gap that remains is whether it is viable to produce A. palmata colonies using assisted fertilization and coral breeding to contribute to the re-establishment of sexually reproducing populations by producing viable offspring.

Here, we describe the results of long-term tracking of A. palmata colonies, produced through assisted reproduction of wild-caught gametes, reared in land-based facilities, then transferred to in-situ nurseries, and finally outplanted onto two reefs in the Mexican Caribbean. This F1 generation of colonies has reached sexual maturity on both reefs and spawned synchronously with wild colonies. An F2 generation of A. palmata juveniles from the outplanted colonies was also produced, providing evidence that colonies raised in the laboratory and outplanted onto reefs have the potential to naturally contribute to the replenishment of genetically-diverse populations, which is critical for restoration programs to be successful over the long term.

Materials & Methods

Gamete collection and larval culture

Acropora palmata gamete bundles (containing eggs and sperm) were collected from wild parent colonies at La Bocana Reef (20.99°N, 86.8°W) in the Puerto Morelos Reef National Park, Mexican Caribbean. Spawning occurred 5 nights after full moon in August 2011 and September 2012. Collecting nets with containers at the top were deployed on presumed genetically distinct colonies by divers. In 2011, spawn was collected from 10 parent colonies and from four parent colonies in 2012. The collecting containers were lidded and taken to the dive boat, where the gamete bundles from all parents were combined for fertilization in 1 µm filtered and UV-C sterilized sea water (FSW). The gametes were gently mixed for 30 s at five-minute intervals to assist the fertilization process and transported to the laboratory at the Unidad Académica de Sistemas Arrecifales of the Universidad Nacional Autónoma de México campus in Puerto Morelos, Quintana Roo, México. After two hours, the mixture was rinsed to remove excess sperm and predators, using fat-separating pitchers, and placed into two 100 L coolers containing FSW maintained at 28 °C until settlement. After four hours, two samples of the mixture were examined under a dissecting microscope at 60X magnification (Motic SMZ-161, Motic Microscopes, Vancouver, Canada), and the number of developing embryos relative to the total number of eggs was determined to estimate fertilization success. Daily 60% water changes were performed for each culture to eliminate unfertilized eggs, metabolized lipids, organic matter, and mucus to reduce bacterial proliferation (Banaszak et al., 2018).

Settlement and post-settlement care in ex-situ nurseries

Substrates in the shape of plugs were made using a 1.5 cm high section of 1″PCV pipe as a mould, using a mixture of Portland cement and beach sand (1:1) and then conditioned in the sea, on pavement in the back reef, for approximately two months prior to larval settlement to promote the growth of coralline algae and biofilm that induce larval settlement and metamorphosis (Negri et al., 2001). The substrates were scrubbed to remove sediment and overgrowing algae, and placed in the culture bins once the larvae began to swim.

Initial settlement was determined at five to eight days post-settlement. At 10 days post-settlement, the F1 settlers were inoculated with symbiotic dinoflagellates freshly isolated from A. palmata fragments using FSW and an air brush (Tytler & Spencer, 1983; Baird et al., 2006). During the inoculation period, the water level in the larval culture bins was reduced and a concentration of 1 × 106 symbiont cells/mL was added. After 16 h, a partial water change was made, and three days later the settlers were inspected for signs of symbiont uptake using a dissecting microscope and fluorescent light.

The F1 settlers were maintained in outdoor aquaria either at UNAM or at Xcaret Aquarium (Figs. 1 and 2). Both ex-situ nurseries were outfitted with flowing sea water at an average temperature of 29 ± 0.6 °C and peak noon irradiance ranging from 300 to 600 µmol m−2 s−1. Temperatures were measured using HOBO water temperature data loggers (HOBO UA-001-64 Pendant Temp) and irradiance was measured using a LI-COR LI-192 underwater quantum sensor connected to a LI-COR 1400 data logger. The settlers were fed daily, initially using commercially available feed (Kent Marine, Franklin, WI, USA) according to the recommendations of the manufacturer, and later with rotifers and fresh Artemia nauplii. The aquarium walls and floors were cleaned twice a week to remove filamentous algae, whereas the substrates were carefully cleaned with the naked eye to reduce algal overgrowth on a weekly basis. Detailed monitoring of the settlers was begun at the third month post-settlement (Table 1). For the 2012 cohort, colony area and number of polyps were obtained from photographs analysed with ImageJ software.

Figure 1 Location of sites where the phases of this study were performed.

(A) Distribution of sites along the Quintana Roo coast. (B) Cuevones Reef where F1 juveniles of A. palmata were outplanted in February 2015 and gravid colonies were recorded in 2019. (C) La Bocana Reef where gametes were collected in 2011 and 2012; UNAM where gametes were fertilized and recruits were bred in an ex-situ nursery; In-situ nursery where juvenile colonies were maintained; and Picudas Reef where outplanted F1 colonies spawned in 2022. (D) Xcaret where an ex-situ nursery was installed and an in-situ nursery was installed at Las Ruinas. Map data ©2023 Google, Landsat/Copernicus, Maxar Technologies.

Figure 2 Timeline of the phases described in this study for the two cohorts of Acropora palmata produced via assisted sexual reproduction.

Acropora palmata colonies were transferred from two ex-situ nurseries (UNAM and Xcaret) to two in-situ nurseries (Puerto Morelos and Las Ruinas) and later to Cuevones Reef and Picudas Reef. The ages of the two cohorts are shown in blue (2011 cohort) and red (2012 cohort) at the top and the corresponding coloured arrows (with N = number of recruits) depict the transfer to the next phase. Samples for fecundity analysis were collected from Cuevones Reef and spawning observations and gamete sampling were undertaken at Picudas Reef. Note: The Las Ruinas in-situ nursery was decommissioned in 2015 when colonies were outplanted to Cuevones Reef and the remaining coloines were transferred to the Puerto Morelos nursery (dashed black arrow).

Table 1 Survival of Acropora palmata cohorts.

The number of colonies of A. palmata produced in 2011 and 2012 that survived, at each time point, in ex-situ facilities, after transfer to an in-situ nursery, and after outplanting onto Cuevones Reef.

	2011 Cohort	2012 Cohort	
	Date	Age	UNAM	XCARET	Date	Age	UNAM	XCARET	
Ex-situ	Sep 2011	1 mo	1155	–	Dec 2012	3 mo	–	393	
	Feb 2012	6 mo	119	35	Jun 2013	9 mo	–	151	
	May 2012	9 mo	12	12	Sep 2013	12 mo	–	112	
	Aug 2012	12 mo	12	12	Nov 2013	14 mo	–	107	
	Sep 2013	25 mo	12	12	Mar 2014	18 mo	–	46	
In-situ	Apr 2014	2.6 y	12	12	Apr 2014	1.6 y	18	18	
	Aug 2014	3 y	12	12	Sep 2014	2 y	18	18	
	Feb 2015	3.5 y	12	12	Feb 2015	2.4 y	18	18	
Reef	Feb 2015	3.5 y	8	12	Feb 2015	2.5 y	5	4	
	Aug 2017	6 y	8	12	Aug 2017	5 y	5	4	
	Aug 2020	9 y	8	12	Aug 2020	8 y	5	4	

Transfer and monitoring of F1 colonies in in-situ nurseries

In April 2014 a subset of colonies from the two cohorts (2011 and 2012) was transferred from the two ex-situ nurseries to two in-situ nurseries (Figs. 1 and 2): one was located in the Puerto Morelos Reef National Park (20.88°N, 87.85°W) at 5 m depth and the other at Las Ruinas (20.42°N, 87.11°W) in Playa del Carmen at 3.8 m depth. The in-situ nurseries consisted of 45 cm high vertical supports made from 2.5 cm diameter PVC tubes incorporated into cement plates that measured 50 cm × 50 cm × 20 cm (L × W × H). Each support had a PVC connector at its free end which was used to mount the coral colony via a compatible connector (Johnson et al., 2011). Each nursery received the same number of juveniles from each cohort: 12 each from the 2011 cohort and 18 each from the 2012 cohort.

The average daily temperature in the Puerto Morelos in-situ nursery, measured by a HOBO water temperature data logger (HOBO UA-001-64 Pendant Temp), was 29 ± 0.5 °C and 28 ± 0.3 °C in the Las Ruinas nursery. Irradiance at solar noon, measured by an Odyssey Waterproof Photosynthetic Active Radiation Logger, ranged between 450 and 600 µmol m−2 s−1 in the Puerto Morelos in-situ nursery and between 230 and 710 µmol m−2 s−1 in the Las Ruinas nursery.

Survival and growth were determined at the same intervals in the Puerto Morelos and Las Ruinas nurseries; for colonies from the 2011 cohort at 2.6, 3, and 3.5 years old, and for colonies from the 2012 cohort at 1.6, 2, and 2.4 years old (Table 1). For growth measurements, the maximum diameter was recorded using a vernier caliper. We compared growth rates between ex-situ and in-situ nurseries via a parametric t-test (RStudio Team, 2020).

Outplanting of F1 colonies onto Cuevones Reef

The lack of natural A. palmata recruitment at Cuevones Reef (21.15°N, 86.74°W) 20 years after a ship grounding (Victoria-Salazar et al., 2023) prompted the selection of Cuevones Reef as an outplant site for A. palmata juveniles. In February 2015, 29 colonies from both the 2011 and 2012 cohorts were harvested from the two in-situ nurseries and outplanted onto Cuevones Reef (Figs. 1 and 2) at 10 m depth. The overall temperature averaged 29 ± 0.5 °C and average peak noon irradiance ranged between 300 and 400 µmol m−2 s−1. Each colony was attached to the reef using a mixture of cement and sand. The colonies were monitored for survival and growth as described in Table 1.

Fertility and fecundity of F1 colonies at Cuevones Reef

In August 2019, tissue samples were collected from the edge of 10 colonies (2011 cohort) and 5 colonies (2012 cohort), placed in ziploc bags with seawater, and transported to the laboratory. The samples were fixed in 10% formaldehyde in FSW for 24 h at 4 °C, then rinsed with FSW. Then, all samples were decalcified by progressively increasing the concentration of hydrochloric acid (HCl, buffered with EDTA, Sodium Tartrate and Potassium) from 1 to 8%. The carbonate-free tissue was preserved in 70% ethanol. Between 10 and 14 polyps were dissected from each colony to document fertility (% polyps with oocytes, Van Veghel & Kahmann, 1994) and fecundity (defined as the number of oocytes per polyp, Kojis & Quinn, 1981 and spermaries per polyp). Polyp density, gonadal index (number of oocytes per cm2), and oocyte diameter were estimated in randomly-selected polyps.

Outplanting of F1 colonies onto Picudas Reef

In September 2020, two colonies from the 2011 cohort and 14 colonies from the 2012 cohort were outplanted onto Picudas Reef (20.88 N, 87.85 W) in the Puerto Morelos Reef National Park (Figs. 1 and 2). Picudas Reef is 3 to 5 m deep, with an overall average temperature of 29 ± 0.2 °C and a peak noon irradiance ranging between 400 and 550 µmol m−2 s−1. The colonies were attached to the reef using a mixture of cement and sand. The survival, growth, and spawning pattern of all 16 colonies were monitored in the summer during the subsequent two years (Fig. 2). The benthic cover of the reef was obtained following the AGRRA method (Lang et al., 2012) at one and 18 months after outplanting.

Spawning of F1 colonies at Picudas Reef

In August 2022, gametes from two of these F1 colonies were collected (Fig. 2) and cross-fertilized at 85 min after spawning using two mL of gamete bundles from each colony. Separately, an extra cross was made 110 min after spawning with one mL of gamete bundles from an unrelated wild colony and 30 mL of the sperm batch from the initial cross of F1 colonies. Embryos were cultured and settled as described above. The F2 settlers were maintained in the UNAM ex-situ nursery under natural solar conditions.

Results

Fertilization and settlement success in ex-situ conditions

Gamete bundles were collected from two of the 10 colonies that spawned in 2011 and all four colonies that spawned in 2012. Fertilization success was 90% in 2011 and 99% in 2012. Approximately 2,000 settlers from the 2011 cohort and 1,000 settlers from the 2012 cohort were inoculated with symbionts and raised in aquaria maintained under natural sunlight. During the first three months post-settlement, survival of the settlers was 95% after which there was a decrease to 31% at 4 months and 0.3% at 18 months for the 2011 cohort (Table 1). Survival of the 2012 cohort was 38% at 9 months and 12% at 18 months. Mortality was due to overgrowth by filamentous algae, failure to take up symbionts despite multiple inoculation attempts, or competition between the coral settlers. In some cases, two or more coral settlers would fuse to form one larger colony. The number of polyps (R2 = 0.96) and colony area (R2 = 0.94) increased exponentially over time in ex-situ culture (Fig. 3). Growth rates in terms of the maximum diameter of the colonies in the ex-situ nursery were 2.5 mm (± 0.7) per month for the 2012 cohort (Fig. 4).

Survival and growth of F1 colonies in the in-situ nursery and after outplanting

The survival of the 60 juvenile colonies placed in the in-situ nursery was 100% for both cohorts during the 10-month period (Table 1). Twenty-nine colonies were outplanted onto Cuevones Reef in the west Coast of Isla Mujeres, Punta Cancún and Punta Nizuc National Park, 16 colonies were outplanted to Picudas Reef in the Puerto Morelos Reef National Park and 15 colonies remained in the in-situ nursery in the Puerto Morelos Reef National Park. Growth rates as determined by following the maximum diameter over time were 1.9 mm ± 0.5 per month just prior to outplanting onto the reef. This was slightly lower but not significantly different (t = 2.16, p = 0.0791) to the growth rates recorded in colonies maintained in the ex-situ nursery during the same period (Fig. 4).

Three and a half years after the 29 colonies were outplanted onto Cuevones Reef, survival of both cohorts was 100%. Growth rates as determined by the maximum diameter increased to 5.6 mm ± 1.8 per month, whereas the growth rates of the 15 colonies from the same cohort that remained in the in-situ nursery were 1.9 mm ± 0.5 per month (Fig. 4).

Prior to outplanting onto Picudas Reef, macroalgal cover was approximately 34% with 23% turf algae, 23% crustose coralline algal and 3% live coral cover. One and a half years after the outplanting the macroalgal cover was 21%, with 27% turf algae, 27% crustose coralline algal and 5% live coral cover. Of the 16 colonies that were outplanted onto Picudas Reef, all colonies remained alive 2.5 years after outplanting. In the same period, the range of growth rates determined by the maximum diameter was wide, from -260 mm to 350 mm per month. The negative growth is presumably due to partial damage to the colonies due to branch breakage from the impact of hurricanes Delta and Zeta that affected the Puerto Morelos Reef National Park in 2020 (Fig. 5).

Figure 3 Growth of Acropora palmata recruits in ex-situ culture during the first 12 months after settlement.

The number of polyps (triangles) and colony area (circles) show an exponential function of recruit age (R2 = 0.9345, y = 0.0975x2.2486). The broken trend line refers to polyp number and the solid line to colony area. Error bars are standard errors of the mean.

Figure 4 Comparison of growth rates of the two cohorts (2011 and 2012) of Acropora palmata F1 colonies.

The colonies were produced by crossing wild-caught gametes and culturing the embryos and larvae through settlement and grow out in an ex-situ nursery. Growth (maximum diameter of the colonies) was measured while in the ex-situ nursery (solid grey bar, n = 24), after transfer to an in-situ nursery (solid white bar, n = 17), and after outplanting onto Cuevones Reef (solid black bars, n2011 = 9, n2012 = 10). Error bars indicate standard deviation of the mean. Means with the same letters are not significant (p > 0.05).

Figure 5 Comparison of growth rates of F1 Acropora palmata colonies outplanted on Picudas Reef.

Growth (maximum diameter of the colonies) was measured in-situ for two years after outplanting at Picudas Reef. Error bars indicate 10th and 90th percentiles of the median. Dashed line denotes zero growth rate.

Fertility and fecundity of outplanted colonies at Cuevones Reef

Based on the dissections of the polyps from the 15 F1 colonies (Figs. 6A and 6B) that were sampled at Cuevones Reef, eight (53%) contained gonads. Of the 96 polyps that were dissected from the eight colonies with gonads, 76% contained oocytes and/or spermaries (Figs. 6C and 6D). Fecundity was 4.03 ± 2.7 (n = 387) oocytes per polyp and there were 2.30 ± 1.5 (n = 221) spermaries per polyp (Fig. 7A). The gonadal index was 63.63 ± 43.35 (n = 8) oocytes cm−2 (Fig. 7B) and 35.03 ± 19.39 (n = 8) spermaries cm−2 (Fig. 7B). The mean maximum diameter of the oocytes (Fig. 7C) was 769 µm ± 141 (n = 42) and the mean diameter of the spermaries (Fig. 7C) was 1.52 mm ± 0.423 (n = 14).

Figure 6 Reproductive capacity of Acropora palmata colonies produced in the laboratory and outplanted onto Cuevones Reef.

(A) A 9-year-old A. palmata colony on Cuevones Reef. Scale bar = 10 cm. (B) Section of an A. palmata colony showing pink-coloured oocytes. Scale bar = one cm. (C) Cross section and (D) longitudinal section of developing oocytes (O) and spermaries (S) in a decalcified coral polyp. Scale bar = one mm. Photo credits: A and B by Sandra Mendoza Quiroz, C and D by Daniela Muñoz Villareal.

Figure 7 Fecundity and gonadal index from eight F1 colonies.

(A) Boxplot of oocytes (black bars) and spermaries (white bars) per polyp. Boxes represent 25th and 75th percentiles and error bars indicate 10th and 90th percentiles. The grey line represents the mean and diamonds indicate outliers. (B) Gonadal index per cm2. (C) Maximum diameter of oocytes (black bar) and spermaries (white bar). Error bars indicate standard deviation of the mean.

Spawning, fertilization, and settlement success from colonies at Picudas Reef

In August 2022, spawning was recorded in 1 colony from the 2011 cohort and 3 colonies from the 2012 cohort (Figs. 8A and 8B, Table 2) and 18 other nearby colonies, 8 wild and 10 colonies of asexual origin, three nights after full moon at Picudas Reef. Two mL of gamete bundles were collected from each of two colonies of the 2012 cohort that spawned. Cross fertilization of these two colonies yielded low fertilization success at 15%. However, when their sperm were crossed separately with bundle gametes from a wild colony, fertilization success increased to 95%. The embryos and larvae produced in the second cross were maintained in ex-situ aquaria with a settlement yield of 2.4% at 13 days after settlement (Fig. 8C). The settled larvae successfully took up symbionts (Fig. 8D). Survival at three months was 1.76%. The average maximum diameter of the settlers was 1.2 ± 0.3 mm (n = 19) and the average area was 1.0 ± 0.4 mm2(n = 19).

Figure 8 Spawning by the F1 generation of Acropora palmata colonies and their offspring.

(A) Spawning by an F1 generation A. palmata colony that was produced by collecting wild gametes, assisting fertilization, culturing embryos and larvae through to settlement, raising to juveniles and outplanting onto Picudas Reef. Scale bar = 10 cm. (B) Closeup of photo A to show setting of gamete bundles in the polyps’ mouths just prior to spawning. Scale bar = five cm. (C) F2 generation of 10-day-old recruits of A. palmata without symbionts and with skeleton clearly visible. Scale bar = one mm. (D) Five-month-old F2 generation A. palmata juvenile in ex-situ culture. Scale bar = one mm. Photo credits: A and B by Esmeralda Pérez Cervantes, C by Sandra Mendoza Quiroz, and D by Victoria Grosso.

Discussion

In this study, we not only show that F1 colonies of Acropora palmata produced by crossing wild-caught gametes can be reared in the laboratory and outplanted onto coral reefs, but that they are reproductively viable and spawn synchronously with wild colonies. Moreover, this is the first report of the production of viable F2 offspring of this species. Thus, the goal to produce potentially self-sustaining breeding populations of this critically endangered species is achievable approximately within a 10-year span.

Both F1 cohorts reared in the laboratory, even after maintenance for extended periods in ex-situ nurseries prior to outplanting onto a degraded site such as Picudas Reef or a damaged area such as Cuevones Reef, released gametes within the known spawning window for this species and on the same night as their wild counterparts. Coinciding in spawning date and time is significant because it means that it is highly probable that fertilization between the laboratory-produced and natural colonies can occur; in fact, we showed that such a combination results in fertilization success of 95%, at least in our laboratory setting.

At seven and eight years old, the outplanted F1 colonies were shown to be fertile, producing both oocytes and spermaries. The size of the oocytes is large at 769 µm ± 141 in diameter, which falls within the values for nine species of Acropora from the Great Barrier Reef that ranged from 601 to 728 µm diameter (Wallace, 1985). The fecundity of the outplanted F1 colonies was 5.61 ± 1.91 oocytes per polyp and falls within the values reported for four species of Acropora from the Great Barrier Reef that ranged from 4.67 ± 0.60 to 6.52 ± 0.84 oocytes per polyp (Pratchett et al., 2019).

The performance of F1 and F2 larvae and settlers was similar to that of other generations raised in our ex-situ larval culture facility (Banaszak et al., 2018; Miller et al., 2021). As shown in this study, reproductively active adult colonies can be produced from wild caught gametes by applying a series of protocols that optimize culture conditions. Intensive care under controlled conditions during the early developmental stages and after settlement can result in colonies that thrive in the reef environment. However, the growth rates of the colonies while in ex-situ nurseries was relatively low compared to the growth rates on the reef, possibly due to limitations associated with cleaning and maintaining corals in ex-situ cultures. The growth rates were higher than those in in-situ nurseries, which may also be influenced by environmental factors such as competition with algae and predation by fish (Mendoza-Quiroz, Pers. Obs.). The growth rates of the colonies once outplanted onto the reef increased approximately three-fold, when compared to their growth rates in ex-situ or in-situ nurseries.

Table 2 Characteristics of the F1 Acropora palmata colonies that spawned in August 2022.

The size, age, and condition of A. palmata colonies as well as the volume of gametes collected from each colony (mL) are shown.

Cohort	Colony ID	Spawn Collection	Gamete volume (mL)	Length (cm)	Width (cm)	Height (cm)	Number of branches	Colony condition	
2011	120	no	∼50 bundles	25	20	30	3	Healthy	
2012	324	yes	2	22	17	15	4	Healthy	
2012	441	yes	2	40	35	45	7	Healthy	
2012	125	yes	0.1	16	18	23	5	Healthy	

The outplanted colonies reached sexual maturity at least within eight years in both Cuevones Reef and Picudas Reef. Low fertilization success (15%) in the crossing of two of the F1 colonies outplanted at Picudas reef indicates that these may be siblings or half-siblings. Both colonies are from the 2012 cohort, when gametes from four parents were crossed. However, fertilization success increased to 95% once the sperm from these two colonies was crossed with eggs from a wild colony.

Recommendations

Based on our experience and the results of this study, we present ten recommendations for restoration practitioners who wish to establish a coral larval propagation program for population replenishment. 1. Collect from as many unrelated colonies as possible to increase fertilization success; 2. Collect from distant colonies due to the high clonality characteristic of this species to ensure maximal genetic diversity; 3. Use equal volumes of gametes from each colony during assisted fertilization; 4. Maintain cultures as clean as possible during fertilization and embryonic and larval development; 5. Use static conditions for early phase culturing; 6. Precondition settlement substrates for a minimum of one month to improve settlement success; 7. Provide intensive care post-settlement by feeding recruits and cleaning substrates of algal overgrowth; 8. Include an intermediate nursery phase; 9. Outplant sexual recruits in a well-spaced pattern and interspersed with unrelated wild colonies to improve fertilization potential and importantly, reduce the chances of inbreeding. If wild colonies are not available, then we suggest that cohorts from different generations and from as many collection sites as possible be intermixed. 10. Undertake long-term monitoring of growth, survival, reproductive capacity and spawning.

Conclusion

Overall, our results show that coral colonies produced using assisted fertilization and ex-situ coral breeding are able to reproduce synchronously with wild populations and thus have the potential to contribute to the re-establishment of self-sustaining, sexually-reproducing populations of Acropora palmata in the Mexican Caribbean. Innovative techniques to scale up these efforts are needed to achieve the long-term goal of re-establishing the structure and function of coral reefs.

Supplemental Information

Supplemental Information 1 Survival, growth, and spawning data

Tthe different parameters we followed during the study.

Click here for additional data file.

Supplemental Information 2 Fertility and fecundity data from Acropora palmata colonies at Cuevones Reef

Click here for additional data file.

Ana Cerón, Rodolfo Raigoza, Rafael Valdéz and Xcaret for access to and use of aquarium facilities. We thank Sergio David Guendulain García, Ana Laura Aguilar Morales, Eduardo Ávila Pech, Gandhi Ramírez Tapia, Esmeralda Pérez Cervantes, and Santiago Zúñiga Cervera for assistance with field work and data collection. We are grateful to M. W. Miller, E. Weil, and two anonymous reviewers for their insightful suggestions that improved the manuscript.

Additional Information and Declarations

Competing Interests

Author Contributions

Data Availability

Anastazia T. Banaszak is an Academic Editor for PeerJ.

Sandra Mendoza Quiroz conceived and designed the experiments, performed the experiments, analyzed the data, prepared figures and/or tables, authored or reviewed drafts of the article, and approved the final draft.

Aurora Urania Beltrán-Torres conceived and designed the experiments, performed the experiments, analyzed the data, prepared figures and/or tables, and approved the final draft.

Maria Victoria Grosso-Becerra conceived and designed the experiments, performed the experiments, analyzed the data, prepared figures and/or tables, authored or reviewed drafts of the article, and approved the final draft.

Daniela Muñoz Villareal conceived and designed the experiments, performed the experiments, analyzed the data, prepared figures and/or tables, authored or reviewed drafts of the article, and approved the final draft.

Raúl Tecalco Rentería conceived and designed the experiments, performed the experiments, analyzed the data, prepared figures and/or tables, and approved the final draft.

Anastazia T. Banaszak conceived and designed the experiments, performed the experiments, analyzed the data, prepared figures and/or tables, authored or reviewed drafts of the article, and approved the final draft.

The following information was supplied regarding data availability:

The raw data are available in Supplemental Files. The data include survival, growth, spawning and settlement data.

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
