# Peer review of "Long-term survival, growth, and reproduction of Acropora palmata sexual recruits outplanted onto Mexican Caribbean reefs"

_PeerJ, doi:10.7717/peerj.15813_

## Round 0.1 · original submission · Major Revisions

Congratulations on completing this body of work. The reviewers all agree that the data in this paper are crucial, and that the study opens the door for more research on coral reproduction and conservation work on A. palmata. Before the manuscript can be published in PeerJ, there are multiple minor issues and one major issue that came up during the review process that needs to be addressed in a major revision. Importantly an issue has been identified with the data presented in Figure 2A, which seems inconsistent with what typically would be found as in number of polyps in a given area. In addition, all reviewers have provided ample feedback on improving the quality of manuscript, specifically in areas where the language used was not clear or redundant. In general, the manuscript needs to be more focused and question driven. One additional minor suggestion: I would like to see in the revised version a small map of the sampling sites. Even though Caribbean researchers in that area are likely familiar with the locations, it would be helpful to include a map to show how far apart the locations are. Also, please report the coordinates in decimal degrees and not degrees, minutes, and seconds to enable automated future data extraction and meta-analysis studies. Please provide a point-by-point rebuttal and make all the requested changes.

Reviewer 1 ·

Basic reporting

Overall, this manuscript presents important information about the rearing, outplanting, and eventual sexual reproduction of F1 Acropora palmata, and initial data on the production of a F2 generation from some of the F1 gametes. This information provides valuable insight into the potential timeline for achieving maturity in sexually-produced coral juveniles used for reef restoration, and shows encouraging survivorship and growth metrics for these endangered species in both ex situ and in situ environments.

For the most part, the manuscript is written clearly, although I have pointed out numerous places where the grammar can be improved. In general, I encourage the authors to read through their manuscript again carefully, with an eye on making the wording more concise and keeping style consistent with reporting data and numbers.

Throughout the manuscript, the in-text citation style is not consistent. The authors have variation in whether they use a comma before the year (i.e. “Miller et al. 2020” vs. “Miller et al., 2020”). Also, they need to always use a “.” in “et al.”, which you they not always do. Please review in-text citations and make sure they follow a consistent style throughout the entire manuscript.

In the introduction section, the authors’ use of references is quite sparse, where actually there is substantial literature on the topics of Acropora population decline, coral restoration efforts to date, and reproductive biology of these species. I encourage the addition of more references to support statements about these subjects.

The final paragraph of the manuscript (lines 286 – 288) does not accurately reflect the findings and main points of the study. Instead, the authors should highlight the successful reproduction of the F1 generation, which is important for achieving the overarching goals of coral restoration (i.e. creating self-sustaining populations). The results presented here do NOT say anything about “enhancing reproductive function” of the population relative to the wild and asexually-produced colonies on the reef, which also spawned when the F1 generation did. I would instead say that the F1 generation can reproduce in synchrony with wild populations, contributing to reproduction on the reef (but not necessarily “enhancing” it like the authors say). I would also avoid claiming that the restoration work here “reestablished the structural function of creating habitat” because the authors do not provide data to back that up (i.e. reef rugosity, fish abundance, etc.), and that is not at all the focus of the study.

Experimental design

This study aligns well with the aims and scope of the journal, and helps to fill a knowledge gap about the survival and reproduction of various generations of corals in restoration efforts. The Methods section is very detailed, and I feel confident that for the most part, this study would be reproducible for other researchers. I especially appreciate the illustrations in Figure 1 and the great photos in Figures 5 and 6 – these are all very helpful for visualization, which is important for many people to help interpret your methods and findings.

One shortcoming – it is not clear whether growth data is derived by adding length, width, and height together (and reporting that sum in mm in Figures 3 and 4), or if the authors are taking whichever of those dimensions is largest and reporting that value in mm.

Validity of the findings

For the most part, the manuscript provides sound, clear data that are well explained and illustrated. However, I would like to see graphs with numerical data on the oocytes, spermaries per polyp, as well as the gonadal index and diameters of oocytes and spermaries. That would be very helpful for visualizing the range of values present in the samples. I do not think that stating these values as text in the Results section is sufficient representation of the data – these should be represented visually.

One issue I have with the final paragraph of the manuscript, as stated above: Lines 286 – 288 do not accurately reflect the findings and main points of the study. Instead, the authors should highlight the successful reproduction of the F1 generation, which is important for achieving the overarching goals of coral restoration (i.e. creating self-sustaining populations). The results presented here do NOT say anything about “enhancing reproductive function” of the population relative to the wild and asexually-produced colonies on the reef, which also spawned when the F1 generation did. I would instead say that the F1 generation can reproduce in synchrony with wild populations, contributing to reproduction on the reef (but not necessarily “enhancing” it like the authors say). I would also avoid claiming that the restoration work here “reestablished the structural function of creating habitat” because the authors do not provide data to back that up (i.e. reef rugosity, fish abundance, etc.), and that is not at all the focus of the study.

Additional comments

Line 23: “restore” spelled incorrectly
Line 24: A. palmata sexual recruits produced from how many parents? Were they batch-combined to create these offspring?
Line 32: “obeyed” does not really make sense in this context – I think “followed” would be better
Line 37: writing “(average +- SD)” after your numbers is not necessary here – this is standard and understood in scientific manuscripts, so you do not need to write it
Line 49: consider rewording, since you had previously used “fecundity” as a quantitative metric and here you are using it qualitatively. I would consider saying instead: “Since colonies of both cohorts were found to be gravid, suggesting they are contributing to the genetic diversity…”
Line 50: Omit one of the uses of “reproducing” in this sentence, since you use it twice in the sentence
Line 54: first paragraph needs indent
Lines 55 – 58: Please add references to support statements about disease outbreaks, low recruitment, and coral restoration interventions
Lines 64 – 66: Please add references to support statements about spawning times, hermaphroditic status, etc.
Lines 69-70: Need to explain why Picudas Reef is an example of coastal development and degradation
Line 78: Should clarify that these outplanted colonies were sexually produced through assisted reproduction
Line 86: Needs indent
Lines 86-87: How many parent colonies were involved in making this batch of offspring? This could be important for understanding and explaining why the 2 colonies later had low fertilization when their gametes were crossed – perhaps they were siblings or half-siblings?
Line 93: Where is the laboratory? Should state the institution(s) involved
Lines 99-102: Consider combining these two sentences to be more concise, as they sort of repeat the same information twice (i.e. “in the form of plugs” and “conditioned” both repeated)
Line 102: Omit “on retrieval from the sea”
Line 125: Needs indent
Line 128: Given that they are 8 months old at this point, I would call them “juveniles” rather than “recruits” (they probably have multiple polyps and have grown)
Line 137: writing “(average +- SD)” after your numbers is not necessary here – this is standard and understood in scientific manuscripts, so you do not need to write it
Line 139: Can you elaborate on what you mean by survival being determined on various levels? Not clear here.
Lines 139-141: It is not clear exactly what you mean by this measurement. Are you adding the values of the length, width, and height together to get the eventual size? Is that what is reflected later in the “growth” data you report in mm in Figures 3 and 4? Need to clarify and explain further.
Line 145: Missing an “and” – should be “and were outplanted”
Line 153: Should be “all”
Line 155: Do not need to capitalize “ethanol”
Line 159: Can you elaborate on what you mean by survival being determined on various levels? Not clear here.
Line 161: This is implied, so you do not need to say that except in figure captions.
Line 166: delete “(average +- SD)”
Line 171: gametes from how many F1 colonies were collected and combined?
Line 174: These should be called “F1 colonies” here, not juveniles, since they are definitely not juveniles anymore in 2022
Line 189: It makes more sense to say “number of polyps”
Line 210: “survival was 100” does not make sense – please consider changing this, and other sentences like it, to something like “All colonies remained alive __ months after outplanting”.
Line 217: “15 colonies of sexual origin” is a confusing way to word this – I suggest saying “the 15 F1 colonies” instead
Lines 220 – 223: Need to employ a consistent scheme for where you put the unit of measure. In some statements here, you say “__ +/- __ unit”, and other times you say “__ unit +/- __” – Please pick one style and use that style throughout the manuscript
Line 239: Delete “the coral species” – Thanks to the introduction we know this is a coral, so it is not necessary to repeat here
Line 239: Again, I think these colonies should be referred to as “F1 colonies” – that is the correct biological term, and is less confusing than “colonies of sexual origin”
Line 243: I would say “is achievable over a ~10 year time span” because yes it is achievable, but your paper is not the first to show that, and indeed it shows this over a longer time span than other studies
Line 244: I would say “both F1 cohorts” rather than both generations, because both the 2011 and 2012 cohorts are the F1 generation
Line 246: “Released” makes more sense than “liberated” here
Line 251: Consider rewording to “At seven and eight years old, the outplanted F1 colonies were shown to be fertile…”
Line 258: Are you referring to larvae and settlers of both the F1 and F2 generations? Please state which generations/cohorts you are specifically referring to
Line 260: Delete “rate” – survivorship does not need the word “rate” after it
Lines 274 – 275: Careful with this statement. Your cleaning protocol does not prove that this helps when corals are outplanted, because you do not have a control group that was not cleaned before outplanting to show any sort of difference. I would avoid or reword this statement here.
Line 276: Reached sexual maturity after how long?
Line 288: Need period at the end of the final sentence.


Table 1: Need to add to the caption what these numbers represent – like “number of F1 colonies surviving at each time point”. As it currently is, it is not clear what these numbers in the table mean.
Figure 3: I suggest adding letters or asterisks to indicate significant differences in growth rate. Also, the different shading patterns of the bars are not explained in the caption or in a legend, so they should be explained.
Figure 4: Please add dashed line at y=0, to show the threshold between negative and positive growth rates. As it currently is, it is very hard to tell if the growth rates ever become positive (which I think they do, but is just not clear with the scale of this y axis)

Annotated reviews are not available for download in order to protect the identity of reviewers who chose to remain anonymous.

·

Basic reporting

This is a manuscript reporting on several years of experimentation with restoration of the foundational scleractinian Acropora palmata on the Caribbean coast of Mexico. The manuscript needs some more information in the introduction to add context for the benefit of the readers and, to enhance the significance of the results and improve the discussion. For example, a short paragraph summarizing why Ap went almost extinct across its entire geographic distribution, and what factors had kept its recovery slow, and the short and long-term results of other Caribbean restoration attempts, why they failed after a few years, why none of them reflects current areas of natural recover of Ap populations. There are many localities where Ap has and is naturally recovering over wide areas much better than any restoration projects.
It is also important to discuss the spatial scales of the restoration projects in comparison with the extensive coral reef systems and some discussion on what are the major differences in the approach used in this project compared to the many other attempts at Ap restoration would enhance the ms.
A couple of paragraphs in the discussion seem to belong in the methods section.
Main conclusions should be clearly stated and maybe put in bullets.

I include suggestions for changes and addition (by line) in the ms.

Experimental design

The experimental design seems to be adequate giving the conditions and the available colonies. There are however some areas where more explanation is needed. For example, how was fertilization success estimated? Why the substrate-plugs needed to be "pre-conditioned".

There are a couple of paragraphs in the discussion (lines 263 - 280) that seem to belong in the methods. Please review.

Validity of the findings

I am curious as why growth rates in the early stages are so low. Is this related to the culture environment? Quantity and quality of light? nutrient availability?

Additional comments

See comments along the ms.

Reviewer 3 ·

Basic reporting

1. There are consistent typos, omitted or extra words, and occasional informal use of language that should be addressed. Some examples are typos in line 23 (‘restore’) and 26 (‘that’ instead of ‘the’), Line 144-145 “were harvested from…were outplanted”, and language such as the use of “no signs of” in Line 76 (consider “no evidence of” for more formal writing). Line 90, use of the word ‘respectively’ not appropriate in this sentence.

2. No clear knowledge gap or research question is identified, and the manuscript feels like a compilation of observations that do not always seem to fit a singular purpose. Please define a research purpose in the introduction (i.e., survival and growth of sexual recruits in mixed nursery rearing, time to reproduction for sexual recruits reared in ex situ nurseries, ability to produce viable offspring with wild colonies). Expand on the specific knowledge gap being filled, or clearly organize around several knowledge gaps.

3. The introduction would benefit from a paragraph about ex situ coral nurseries and reference previous articles with survival in ex situ culture and after outplanting from ex situ nurseries. One of the strengths of this paper is the high survival post-outplant and long-term monitoring data, so this should be highlighted.

Experimental design

1. Although the manuscript does not follow a perfect experimental design, the reviewer recognizes the extreme difficulty of the work described and the long-term compilation of data.

2. The data is presented in a way that seems as if most of it was opportunistically collected and compiled, and it is worth noting that this is the culmination of over a decade of work and presents relatively rare long-term data on the success of sexually derived corals raised ex situ and the onset of sexual reproduction.

3. Some parts of the Materials and Methods could benefit from additional detail or clarity. For example, the authors state that the “substrate plugs were manufactured”. Were they handmade by the authors using molds or purchased from a vendor with a Brand/Manufacturer that could be referenced? To what micron level is the FSW filtered?

4. Please check the units in Figure 2A. It seems like a colony that is 18 square centimeters should have far more than ~60 polyps. Looking at the raw data, I think this error is coming in the conversion of square millimeters (original measurement unit) to square centimeters, as the decimal place actually needs to be moved two places to convert area.

5. Similar concern over line 236 with an area of 1.49 square centimeters yet the mean diameter was only 1.2 mm.

6. With regard to the AGRRA surveys, the author states that the data is not shown (Line 208) so consider not discussing algal cover as this is likely not due to the outplant of corals and could have many other factors not controlled for or well replicated (such as seasonal variation). Consider limiting to the live coral cover metric only.

Validity of the findings

1. This work represents a long-term study on the success of assisted sexual reproduction for coral restoration and the successful use of multiple nursery types in recruit rearing. The results are scientifically sound, with the possible exception of area measurements mentioned previously which should be addressed. The results are strengthened by multiple years and locations for spawning, rearing, and outplanting. The results support the use of similar techniques in coral restoration and managed breeding activities.

2. The conclusions could also benefit from additional clarity around a target research question and specific knowledge gap, similar to the introduction. This could be the ability for sexual recruits to spawn with wild colonies, and/or the use of mixed-strategy nursery rearing and grow-out for sexual recruits, resulting in high post-outplant survival.

Additional comments

The manuscript should be given an additional thorough edit for typos and language use. It also needs a review of all area units with conversion of square mm to square cm, in analysis and figures.

This is a unique and complex long-term project that is important to the coral restoration and spawning community.

Overall, the data presented is scientifically sound yet some of the data may not be directly relevant to a defined question. The authors could consider removing some aspects (such as correlation between number of polyps and colony area) that are not directly relevant to the question at hand, once that question is better defined.

---

## Round 0.2 · Minor Revisions

Thank you for making the changes to the manuscript, and congratulations again on completing this much-needed study. The reviewers agree that the manuscript is ready for publication. I am sending it back for minor revision to give you the opportunity to fix the few typos and address the minor comments that were raised in this version before it is published.

Reviewer 1 ·

Basic reporting

The authors have done well to make the language clearer and more concise throughout the manuscript, thus making the updated version read nicely. They have added more references to the Introduction, which has helped make this section more comprehensive and better contextualize the study in the literature. Figures and tables are professional, and raw data have been shared. The only minor comment I have is that the study still does not feel like it has a driving question that is presented in the beginning to drive the narrative, though I do think it does well to answer whether and in what time scales A. palmata can be used for sexual restoration efforts.

Experimental design

Thank you very much for adding the quantitative data on gametes per polyp, gonadal index, and diameter of oocytes and spermaries. These data add additional rigor to your findings, and contribute useful numbers against which others in the field may compare their own values in the future. The methods have been described in sufficient detail to replicate. I do not feel that a research question was well defined, but rather that a series of actions are being described (that were not necessarily driven by anything specific).

Validity of the findings

Perhaps could add something about the benefits this study presents to the literature and the knowledge gap being addressed, in addition to recommendations for restoration practitioners.
With the revisions, this manuscript presents thorough, clear conclusions that are backed up by the data and now adequately limited to supporting their results. The authors have done well to reframe some of their initial claims, and ensure that conclusions are drawn from data and observations reported here without additional extrapolation.

Additional comments

Line 48: Gardner misspelled
Line 270: do not need comma between “gametes” and “can”

Overall, I am very happy with how thoroughly the authors have addressed each comment/suggestion I made in the first round of revisions. The resulting, updated manuscript reads well, and presents a more polished picture of the study for readers. Well done!

Reviewer 3 ·

Basic reporting

The edited version of the manuscript is significantly improved in it's lack of typos and grammatical errors, and more concise and consistent language. Overall readability is significantly better. Great job.

Improvements to the introduction and conclusions outline the purpose more clearly.

Experimental design

The added detail to the methods section improves this section and will be very helpful to other researchers trying to replicate any portion of the work.

The errors is unit conversion have been addressed or figures removed.

Validity of the findings

The conclusions and the introduction now outline a much clearer purpose, and removal of some ancillary data/figures and addition of more relevant figures, photographs and data suggested by other reviewers make this a significantly improved manuscript.

Additional comments

No additional comments

---

## Round 0.3 · accepted · Accept

Thanks for making all the requested minor changes. The manuscript is now accepted for publication. PeerJ staff will be in contact with you regarding the next steps.